# Acidocalcisome-like vacuoles constitute a feedback-controlled phosphate buffering system for the cytosol

**Samuel Bru[1,2], Lydie Michaillat Mayer[1], Geun-Don Kim[1], Danye Qiu[3], Henning J Jessen[3], Andreas Mayer[1]***

[1]Département d'immunobiologie, Université de Lausanne, Epalinges, Switzerland; [2]Department of Biomedical Sciences, Faculty of Medicine and Health Sciences, Universitat Internacional de Catalunya, Barcelona, Spain; [3]Institute of Organic Chemistry University of Freiburg and CIBSS – Centre for Integrative Biological Signalling Studies, University of Freiburg, Freiburg, Germany

## eLife Assessment

This study provides **fundamental** insights into eukaryotic phosphate homeostasis by demonstrating how yeast vacuoles dynamically regulate cytosolic phosphate levels. The conclusions are **convincing**, supported by an elegant combination of in vitro assays and in vivo measurements. This study will be of interest to cell biologists, particularly for those who are working in the field of phosphate metabolism.

*For correspondence: andreas.mayer@unil.ch

Competing interest: The authors declare that no competing interests exist.

**Abstract** Cells experience strong variations in the consumption and availability of inorganic phosphate ($P_i$). Since $P_i$ is an essential macronutrient but excess $P_i$ has negative impacts on nucleotide hydrolysis and metabolism, its concentration must be maintained in a suitable range. Conserved storage organelles, acidocalcisomes, provide this buffering function. We used acidocalcisome-like yeast vacuoles to study how such organelles are set up to perform this task. Our combined in vitro and in vivo analyses revealed that their ATP-driven polyphosphate polymerase VTC converts cytosolic $P_i$ into inorganic polyphosphates (polyP), which it transfers into the vacuole lumen. Luminal polyphosphatases immediately hydrolyse this polyP to establish a growing reservoir of vacuolar $P_i$. Product inhibition by this $P_i$ pool silences the polyphosphatases, caps $P_i$ accumulation, and favours vacuolar polyP storage. Upon cytosolic $P_i$ scarcity, the declining inositol pyrophosphate levels activate the vacuolar $P_i$ exporter Pho91 to replenish cytosolic $P_i$. In this way, acidocalcisome-like vacuoles constitute a feedback-regulated buffering system for cytosolic $P_i$, which the cells can switch between $P_i$ accumulation, $P_i$ release, and high-capacity phosphate storage through polyP.

## Introduction

Cells actively manage the concentration of $P_i$ in their cytosol because they must strike a balance between conflicting goals (*Austin and Mayer, 2020*). On the one hand, $P_i$ is a product of nucleotide-hydrolysing reactions. Its concentration has a significant impact on the free energy that these reactions can provide to drive metabolism. Since, for this reason, excessive $P_i$ concentrations might stall metabolism, we can expect that cytosolic $P_i$ must remain limited. But $P_i$ is also an essential macronutrient. As a major constituent of nucleic acids and phospholipids, and as an important modifier of proteins, carbohydrates and many metabolites, it is consumed in large quantities for anabolic reactions. This can lead to situations where progression of the S-phase of the cell cycle is limited by the $P_i$ uptake

capacity of the cells (*Bru et al., 2017*; *Bru et al., 2016*; *Gillies et al., 1981*; *Neef and Kladde, 2003*). Rapidly dividing cells, such as yeast cells, remedy this problem by maintaining phosphate stores in the form of inorganic polyP. PolyPs are chains of $P_i$ linked through phosphoric anhydride bonds, which can be stored in acidocalcisome-like organelles (*Okorokov et al., 1980*; *Urech et al., 1978*). Acidocalcisomes are lysosome-related organelles that have been characterised by pioneering work mainly in trypanosomes, but they have been isolated from a multitude of other organisms (*Docampo et al., 2005*). Their defining chemical features are their acidity, high concentrations of polyP, Ca, Mg, Zn, Fe, K, Na, and basic amino acids, and they are implicated in osmoregulation and the homeostasis of phosphate and metals (*Docampo, 2024*; *Docampo et al., 2005*; *Girard-Dias et al., 2023*). Yeast vacuoles share all these properties and are hence good and easily accessible models for studying aspects of acidocalcisomal mechanisms, such as the synthesis and turnover of polyP (*Austin and Mayer, 2020*; *Dürr et al., 1979*; *Huber-Wälchli and Wiemken, 1979*; *Urech et al., 1978*) or metal uptake (*Klompmaker et al., 2017*). But they have an additional, lysosome-like face, enabling them to assume also hydrolytic degradative functions for the cells (*Li and Kane, 2009*). Therefore, we refer to them as acidocalcisome-like organelles.

The polyP stores of vacuoles are accessed by cells when $P_i$ becomes limiting in the environment, or when they face a sudden increase in $P_i$ consumption, e.g., during metabolic transitions from respiratory growth to fermentation, which requires many more phosphorylated metabolites (*Gillies et al., 1981*; *Nicolay et al., 1983*; *Nicolay et al., 1982*; *Shirahama et al., 1996*; *Thomas and O'Shea, 2005*). In such cases, cellular polyphosphate content decreases. It has therefore been proposed that polyPs are re-converted into $P_i$ to buffer shortages in cytosolic $P_i$ (*Nicolay et al., 1983*; *Nicolay et al., 1982*). While this hypothesis is straightforward, we are lacking a coherent concept of how a $P_i$ buffer based on an acidocalcisome-like organelle might work. Pioneering work on these organelles has revealed several characteristic and conserved features, such as their acidity and their high content of basic amino acids and divalent cations (*Docampo, 2024*; *Docampo and Huang, 2016*). Of direct relevance to $P_i$ homeostasis is the high capacity of acidocalcisome-like organelles for storing polyP. PolyP can greatly vary in length, from two to hundreds of phosphate units. The membrane of acidocalcisome-like organelles can carry polyP polymerases, such as VTC, and $P_i$ transporters, such as Pho91 (*Gerasimaitė and Mayer, 2016*; *Huang and Docampo, 2015*; *Hürlimann et al., 2007*; *Jimenez and Docampo, 2015*; *Müller et al., 2002*; *Müller et al., 2003*; *Wang et al., 2015*). VTC is a coupled polyP polymerase and translocase (*Gerasimaitė et al., 2014*), which synthesises polyP by transferring the γ-phosphate of cytosolic ATP onto an elongating polyP chain (*Hothorn et al., 2009*). PolyP is generated by a catalytic domain in the centre of the Vtc4 subunit of the complex (*Hothorn et al., 2009*). The activity of this subunit is controlled through the SPX domains of VTC (*Wild et al., 2016*), which may associate into a dimerised inactive state (*Pipercevic et al., 2023*). They can be released from this state through the action of inositol pyrophosphates, signalling molecules that accumulate when $P_i$ is abundant in the cytosol and activate $P_i$ storage in the form of polyP. The $P_i$ state of the cytosol is communicated to VTC through a specific inositol pyrophosphate, $1,5\text{-}IP_8$ (*Chabert et al., 2023*; *Gerasimaitė et al., 2017*; *Kim et al., 2025*; *Kim et al., 2023*).

The postulated direct channelling of polyP from the site of its synthesis through the vacuolar membrane (*Gerasimaitė et al., 2014*) is facilitated by the structure of the VTC complex. The catalytic domain of VTC has its exit for polyP right at the entry of a polyP-conducting channel, which is formed through the transmembrane domains of the VTC complex itself (*Gerasimaitė et al., 2014*; *Guan et al., 2023*; *Liu et al., 2023*; *Müller et al., 2002*; *Müller et al., 2003*). This channel was proposed to exist in an open and closed conformation and to be gated through polyP (*Liu et al., 2023*). PolyP translocation may be driven by the electrochemical potential across the membrane, which could move the highly negatively charged polyP chain through electrophoresis. This can explain why efficient polyP synthesis depends on proton pumps such as the V-ATPase or $H^+$-pumping pyrophosphatases (*Freimoser et al., 2006*; *Gerasimaitė et al., 2014*; *Wild et al., 2016*; *Lemercier et al., 2004*).

The transporter Pho91 was proposed to export vacuolar $P_i$ (*Hürlimann et al., 2007*). Several properties of yeast Pho91 and its homologues from other organisms are consistent with this view. Its ablation increases vacuolar $P_i$ and polyP content, in yeast, as well as in trypanosomes (*Farofonova et al., 2023*; *Hürlimann et al., 2007*; *Jimenez and Docampo, 2015*), and it weakly induces the phosphate starvation programme in yeast (*Pinson et al., 2004*), suggesting that it may induce cytosolic $P_i$ scarcity. Furthermore, patch-clamp analyses of Pho91-like channels from yeast, trypanosomes, and plants

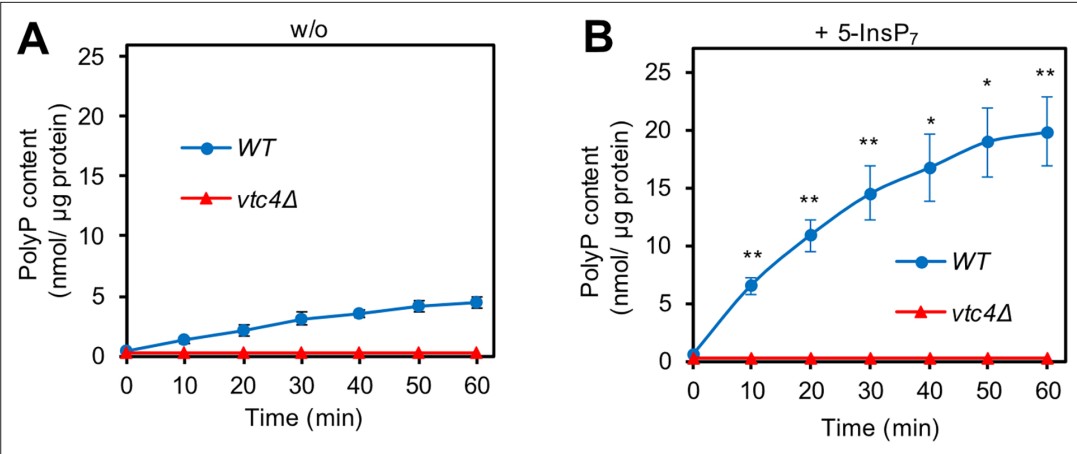

**Figure 1.** VTC- and 5-IP$_7$-dependent polyphosphate (polyP) synthesis by isolated vacuoles. Vacuoles were isolated from logarithmic cultures of BY4742 wildtype cells (WT) or from isogenic vtc4Δ cell strains. They were incubated in polyphosphate synthesis assays without (**A**) or (**B**) in the presence of 50 µM 5-InsP$_7$. At indicated times, aliquots were withdrawn, solubilised in Triton X-100, and polyphosphate was quantified through the polyP-dependent fluorescence of added DAPI. Means ± SD (standard deviation) of three independent experiments are shown. *p<0.05, **p<0.005 for a paired t-test comparing WT with and without 5-InsP$_7$ at each time point.

have shown their P$_i$ permeability and the dependence of their directionality on a pH gradient across the membrane (*Potapenko et al., 2019*; *Potapenko et al., 2018*; *Wang et al., 2015*). In the presence of a pH gradient, the Pho91 homologue from rice, OsSpx-MFS3, mediates P$_i$ flux along the proton gradient, consistent with a function in exporting P$_i$ from the acidic lumen of vacuoles (*Wang et al., 2015*) towards the cytosol.

Acidocalcisome-like organelles also contain polyphosphatases in their lumen (*Gerasimaitė and Mayer, 2016*; *Kulakovskaya et al., 2021*; *Lander et al., 2016*; *McCarthy and Downey, 2023*), which can convert polyP back into P$_i$. In baker's yeast, two vacuolar polyphosphatases are known: Ppn1 and Ppn2 (*Andreeva et al., 2019*; *Gerasimaitė and Mayer, 2017*; *Sethuraman et al., 2001*). This means that a chain of polyP, when being synthesised by VTC and arriving in the vacuolar lumen, is immediately exposed to hydrolytic enzymes that will degrade it. While this seems at first sight paradoxical, we explored the hypothesis that the co-existence of polyP-synthesising and polyP-hydrolysing activities might be a key feature conveying to acidocalcisome-like organelles the capacity to buffer cytosolic P$_i$. That these organelles have a critical role to play in this process is illustrated by observations in yeast, where artificial up- or downregulation of vacuolar polyP synthesis suffices to drive the cytosol into a state of P$_i$ starvation or P$_i$ excess, respectively (*Desfougères et al., 2016a*). Furthermore, the presence of polyP reserves delays the activation of the transcriptional phosphate starvation response, the PHO pathway (*Thomas and O'Shea, 2005*). We hence explored the capacity of isolated yeast vacuoles to interconvert polyP and P$_i$, and we characterised the roles played by the vacuolar polyphosphatases Ppn1 and Ppn2 and the vacuolar P$_i$ transporter Pho91. Our observations can be combined with previous findings to yield a coherent model of how an acidocalcisome-like organelle can operate as a P$_i$ buffer for the cytosol.

## Results

We explored the interplay of VTC, polyphosphatases, and Pho91 in the accumulation of polyP and P$_i$ inside vacuoles using an in vitro system with purified organelles. Vacuoles can be isolated in intact form when the cells are gently opened by enzymatic digestion of the cell wall and disruption of the cell membrane by low concentrations of DEAE-dextran (*Dürr et al., 1975*). Organelles isolated in this way can perform many of their normal cellular functions, such as membrane fusion, membrane fission, and autophagy (*D'Agostino and Mayer, 2019*; *Kunz et al., 2004*; *Michaillat et al., 2012*; *Sattler and Mayer, 2000*). They also contain active polyP polymerase (VTC) and active polyphosphatases (Ppn1 and Ppn2) and they can synthesise and import polyP (*Gerasimaitė et al., 2014*; *Gerasimaitė et al., 2017*).

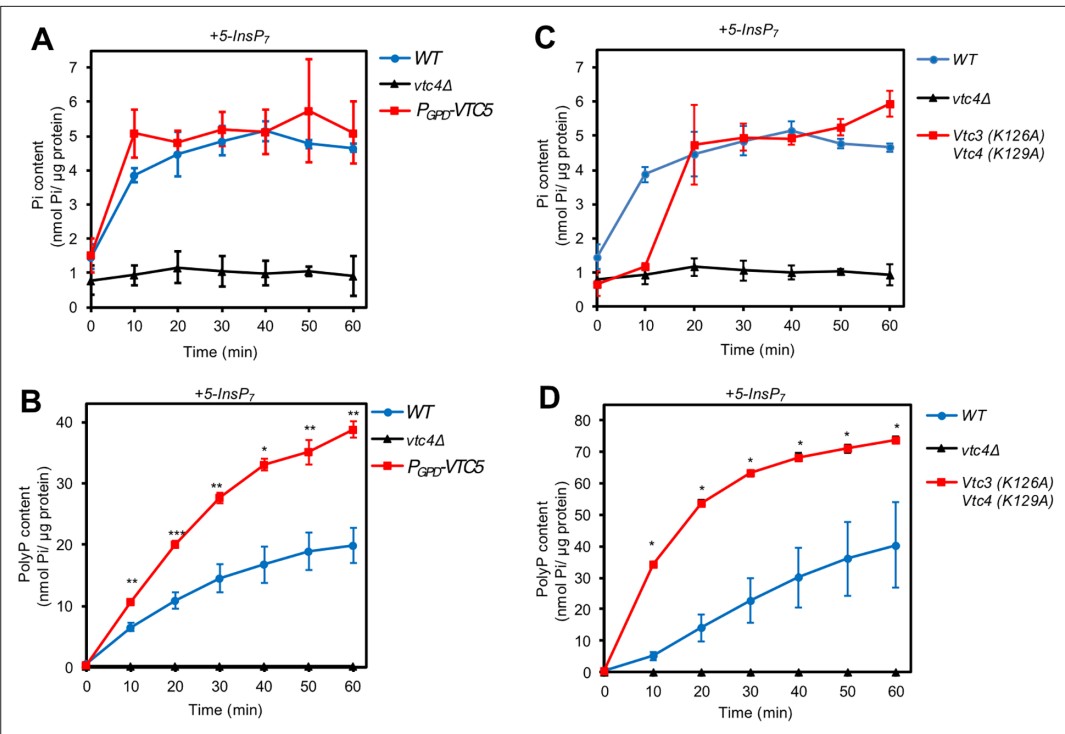

**Figure 2.** VTC-dependent accumulation of $P_i$ in isolated vacuoles. Vacuoles were isolated from the indicated logarithmically growing BY4742 (WT) cells. Polyphosphates (polyP) overproduction was achieved (**A, B**) either through overexpressing VTC5 from the strong GPD promoter, or (**C, D**) by expressing the hyperactivating vtc3[k126A] and vtc4[K129A] alleles from their native promoters as the sole source of these two proteins. The vacuoles were incubated with an ATP-regenerating system and 50 µM 5-InsP$_7$ under conditions allowing polyP synthesis. At indicated times, the vacuoles were solubilised with Triton X-100 and $P_i$ was assayed through malachite green. Graphs represent the mean ± SEM of three independent experiments for each strain. *$p<0.05$, **$p<0.005$, ***$p<0.001$ from a paired t-test comparing each time WT vs VTC5 overexpression or vtc3[k126A] vtc4[K129A].

The online version of this article includes the following figure supplement(s) for figure 2:

**Figure supplement 1.** Isolated vacuoles during polyphosphate (polyP) synthesis.

## Vacuolar $P_i$ accumulation depends on polyP synthesis

Purified vacuoles were incubated with an ATP-regenerating system under conditions that allow these organelles to synthesise polyP in vitro (*Gerasimaitė et al., 2014*). After different periods of incubation, the organelles were sedimented and solubilised in detergent. Vacuolar accumulation of polyP was assayed through DAPI fluorescence, and $P_i$ was measured through malachite green. VTC is stimulated by a variety of inositol pyrophosphates, most efficiently by 1,5-InsP$_8$ (*Gerasimaitė et al., 2017*; *Gerasimaitė et al., 2014*). Since 1,5-InsP$_8$ is not commercially available, we used saturating concentrations of 5-InsP$_7$ for our experiments (*Pavlovic et al., 2015*). This compound stimulates VTC with a higher EC$_{50}$ but to the same maximal activity as 1,5-InsP$_8$, and it was more readily available to us for routine experiments. Under the conditions used here, the organelles rapidly and efficiently produced polyP (*Figure 1*), in line with previous results (*Gerasimaitė et al., 2017*; *Gerasimaitė et al., 2014*). This production was stimulated by 5-InsP$_7$, reaching 0.7 nmol of phosphate units per µg of vacuolar protein per min. This means that already within 10 min, wildtype vacuoles produced a mass of polyP equivalent to their total protein content, indicating how efficiently these organelles synthesise polyP. This signal was entirely dependent on VTC, because it was not observed in a mutant lacking the catalytic subunit of VTC (*vtc4Δ*).

Isolated wildtype vacuoles also accumulated $P_i$ with significant efficiency, at an initial rate of at least 0.3 nmol/µg/min (*Figure 2A*). Vacuoles from *vtc4Δ* cells showed only 8% of this $P_i$ signal. Most of it was not time-dependent and hence may represent a background signal from the organelle preparation. Thus, vacuolar $P_i$ accumulation depended on polyP synthesis through VTC. But the kinetics of polyP

and $P_i$ accumulation were remarkably different. Whereas polyP continued to accumulate throughout the entire incubation period, $P_i$ accumulated rapidly in the initial phase but reached a plateau within 30 min (*Figure 2A and B*). Thus, although polyP production was essential for $P_i$ accumulation, it could not become the limiting factor for it in the second phase. This is further illustrated by $P_i$ accumulation in strains showing enhanced polyP synthesis, such as overexpressors of VTC5 (*Figure 2A and B*) or strains carrying hyperactivating substitutions in the SPX domains of VTC (*vtc3$^{K126A}$ vtc4$^{K129A}$*; *Figure 2C and D*; *Desfougères et al., 2016b*; *Wild et al., 2016*). Vacuoles from these strains accumulated polyP at a two to seven times higher initial rate than the wildtype and to higher concentrations. Nevertheless, their accumulation of $P_i$ arrested at the same level as that of wildtype vacuoles, at 5 nmol/µg vacuolar protein. To estimate the corresponding luminal concentration of $P_i$, we measured the diameters of 100 isolated wildtype vacuoles and counted the number of vacuoles per µg of vacuolar protein. With an average diameter of 0.8 µm (*Figure 2—figure supplement 1*) and $5*10^7$ vacuoles/µg of protein, we can estimate that $P_i$ accumulation in vacuoles incubated with 5-InsP$_7$ reached a plateau at a luminal concentration of around 30 mM. *vtc4Δ* vacuoles showed only a very small increase of less than 3 mM in luminal $P_i$ content. The 30 mM of $P_i$ accumulating in wildtype vacuoles is a substantial concentration, which is in the range of the in vivo $P_i$ concentration of 25 mM that was measured by $^{31}$P-NMR spectroscopy of yeast cells under conditions where vacuolar $P_i$ dominates the signal (*Okorokov et al., 1980*). Since, upon $P_i$ scarcity, cytosolic $P_i$ drops to 1 mM (*Okorokov et al., 1980*), a substantial $P_i$ concentration gradient across the vacuolar membrane could drive rapid replenishment of the cytosol from a readily accessible vacuolar pool of $P_i$.

## Vacuolar $P_i$ accumulation is limited by feedback inhibition of Ppn1 and Ppn2

We asked why vacuolar $P_i$ accumulation quickly forms a plateau whereas vacuolar polyP continues to accumulate. To this end, we explored the hypothesis that vacuolar polyphosphatases become inhibited when their product, $P_i$, has accumulated. To test polyphosphatase activity, we liberated the polyphosphatases from isolated vacuoles by detergent lysis, followed by incubation with synthetic polyP as a substrate and DAPI as an indicator for polyP. PolyP was efficiently degraded by this extract. The apparent polyPase activity was significantly reduced by 3 mM $P_i$ and efficiently silenced by 30 mM $P_i$ (*Figure 3A*). To differentiate the contributions of Ppn1 and Ppn2, we also analysed the respective deletion mutants and performed the assay at higher polyP concentration in the presence of $Mg^{2+}$, which favours Ppn1 activity, or in the presence of $Zn^{2+}$, which supports activity of Ppn1 and Ppn2 (*Gerasimaitė et al., 2017*). In the presence of $Zn^{2+}$, the substrate was consumed in less than 3 min (*Figure 3*). Degradation was delayed in vacuoles from *ppn1Δ* or *ppn2Δ* mutants, and it was suppressed in vacuoles from a *ppn1Δ ppn2Δ* mutant, in which both polyphosphatases were ablated. In incubations with only $Mg^{2+}$ instead of $Zn^{2+}$, which stimulates the activity of Ppn1 much more than that of Ppn2, polyP degradation was slower and genetic ablation of Ppn1 sufficed to stabilise polyP. The addition of 30 mM potassium phosphate, which is equivalent to the maximal $P_i$ accumulation in vacuoles that we observed in *Figure 2*, attenuated degradation of polyP 5- to 10-fold, both through Ppn1 (assayed in *ppn2Δ* and in samples without $Zn^{2+}$) and through Ppn2 (assayed in *ppn1Δ*). That both polyphosphatases are inhibited by $P_i$ at this concentration is consistent with the notion that product inhibition of Ppn1 and Ppn2 might limit the conversion of polyP into $P_i$ in the vacuolar lumen and thus define the maximal concentration of the vacuolar $P_i$ reservoir.

## The transporter Pho91 limits vacuolar $P_i$ accumulation in an inositol pyrophosphate-dependent manner

To allow vacuoles to function as a $P_i$ buffer for the cytosol, the $P_i$ in the vacuolar lumen should become accessible in a regulated manner. The $P_i$ transporter Pho91 is a prime candidate for mediating regulatable efflux because it is regulated by InsPPs through its SPX domain (*Hürlimann et al., 2007*; *Potapenko et al., 2019*; *Potapenko et al., 2018*; *Wang et al., 2015*). However, its vacuolar localisation is challenged by reports that yeast can show significant growth mediated by Pho91 as the sole Pi transporter, and that, upon overexpression, Pho91 rescues growth to a similar degree as any of the four yeast plasma membrane $P_i$ transporters, Pho84, Pho87, Pho89, and Pho90 (*Wykoff et al., 2007*; *Wykoff and O'Shea, 2001*). Since this capacity to feed the cells is difficult to reconcile with a vacuolar localisation – unless the cells would absorb $P_i$ through fluid phase endocytosis and subsequent

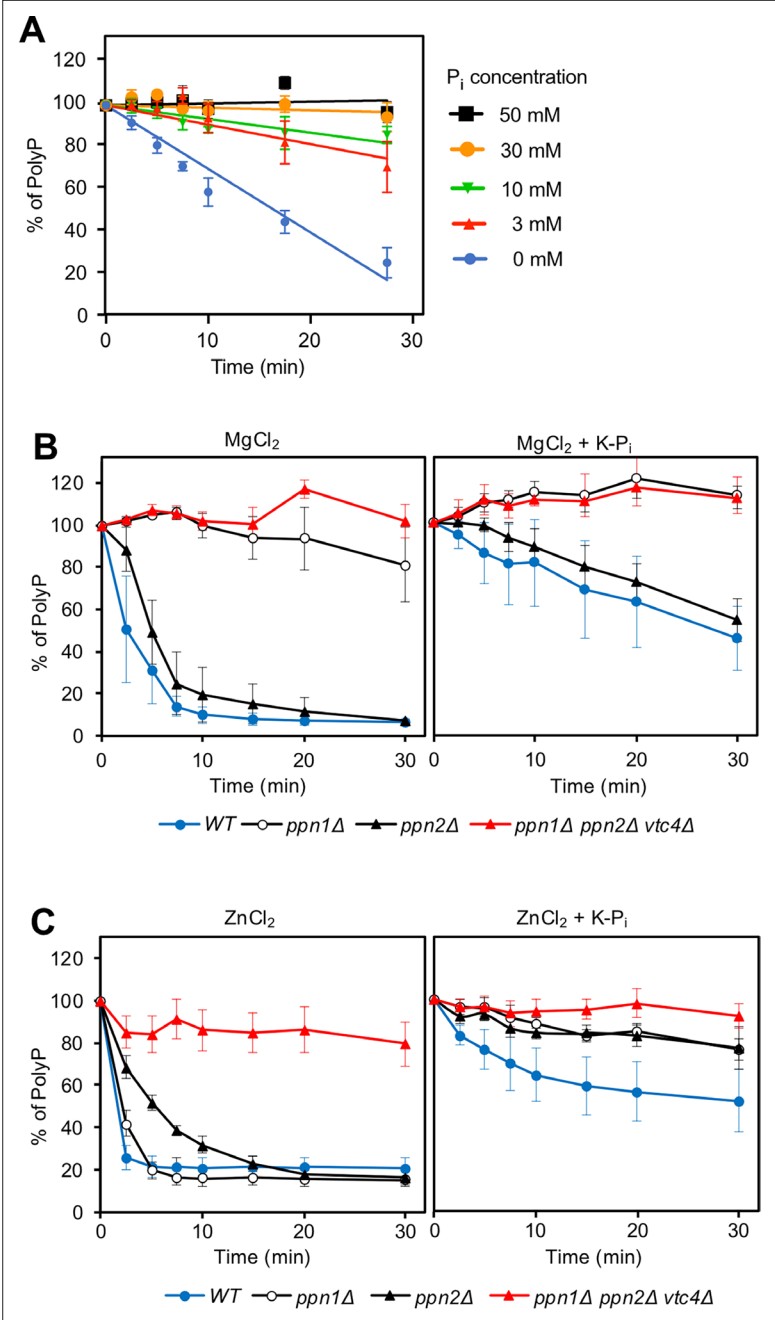

**Figure 3.** Effect of $P_i$ on polyphosphatase activity. (**A**) $P_i$ titration. Vacuoles were isolated from logarithmically growing wildtype cells (BY4742). The organelles were diluted in polyphosphatase assay reaction buffer, which contained 0.1% Triton X-100 and hence liberated the luminal polyphosphatases. This lysate was incubated with 30 µM polyP$_{300}$ as a substrate and supplemented with 1 mM ZnCl$_2$ and the indicated concentrations of K-P$_i$ pH 6.8. After the indicated times of incubation, the remaining polyP was quantified through DAPI. The DAPI signal at the beginning of the incubation served as 100% reference. Graphs represent the means ± SEM of three independent experiments. (**B, C**) Differentiation of $P_i$ effects on Ppn1 and Ppn2. Vacuoles were isolated from the indicated logarithmically growing strains, lysed and used in polyP degradation assays with 300 µM polyP$_{300}$ as in (**A**). These assays were performed in the presence or absence of 30 mM K-P$_i$ pH 6.8 and 1 mM MgCl$_2$ (**B**) or, instead of this, with 1 mM ZnCl$_2$ (**C**) as cation supporting catalytic activity.

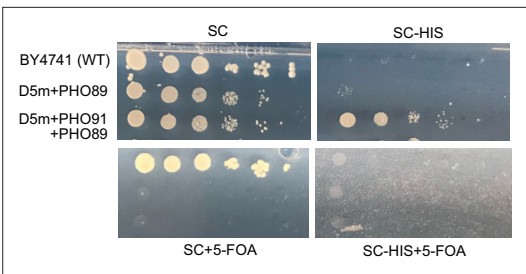

**Figure 4.** Pho91 cannot replace other P$_i$ transporters to support growth of yeast. We generated a BY4741 strain with a quintuple deletion of the genes for the known P$_i$ transporters, PHO84, PHO87, PHO89, PHO90, and PHO91 (D5m). These cells were kept alive by expressing the gene for the plasma membrane P$_i$ transporter PHO89 from a URA3-based centromeric (single copy) plasmid (pRS416). PHO91 was expressed from a HIS3-based centromeric plasmid (pRS315). Cells were plated in a dilution series on synthetic complete (SC) lacking histidine (SC-HIS) to verify that the cells had the HIS3-based PHO91 plasmid, or on SC with 5-fluoro-orotic acid (5-FOA), a drug that forces cells to lose the URA3-based pRS416 and thus to live without PHO89. PHO91 as the sole P$_i$ transporter gene (on SC + 5-FOA) does not allow cells to grow.

The online version of this article includes the following figure supplement(s) for figure 4:

**Figure supplement 1.** Effects of N- and C-terminal fluorescent protein tags on the localisation of Pho91.

export from the vacuole (***Klompmaker et al., 2017***) – we reinvestigated this aspect. Instead of relying on a downregulatable PHO84 background, which results in a low but constitutive background expression of the potent P$_i$ importer Pho84 in addition to Pho91 (***Wykoff et al., 2007***; ***Wykoff and O'Shea, 2001***), we took a more stringent approach, using plasmid shuffling to completely remove PHO89 when PHO91 is brought in. To this end, we generated a quintuple knockout strain lacking these five P$_i$ transporters (Pho84, Pho87, Pho89, Pho90, Pho91), which was kept alive through expression of Pho89 from a URA-based plasmid. Plasmid shuffling allowed to exchange this plasmid against others expressing an individual P$_i$ transporter. Whereas an individual plasma membrane P$_i$ transporter such as Pho89 supported normal colony formation, confirming previous observations (***Wykoff et al., 2007***; ***Wykoff and O'Shea, 2001***), we could not confirm that Pho91 supports slower growth. In our plasmid shuffling approach, Pho91 did not support growth at all (***Figure 4***).

An overexpressed GFP-Pho91 fusion had been reported to localise to vacuoles (***Elbaz-Alon et al., 2014***; ***Hürlimann et al., 2007***), a phenotype that we confirmed (not shown). Since overexpression of membrane proteins in yeast easily leads to their erroneous accumulation in vacuoles, we re-investigated Pho91 localisation in the absence of a strong, overexpressing promoter. When we expressed C- or N-terminal GFP fusions of Pho91 from the authentic PHO91 promoter, we observed considerably weaker signals and different and complex localisation patterns (see ***Figure 4—figure supplement 1*** for examples). These localisation patterns depended on the nature of the fluorescent tag and the linker peptides used to attach it to Pho91, and they often showed significant accumulation in the ER. This suggests that N- and C-terminal fluorescent protein tags interfere with intracellular trafficking of Pho91.

We hence used mass spectrometry of purified vacuoles to test whether the non-tagged, endogenous Pho91 is a vacuolar protein. To this end, we determined the enrichment of proteins in the vacuolar fraction relative to a whole cell extract (***Supplementary file 1***). Peptides from Pho91 were enriched in the purified vacuoles to a similar degree (39-fold) as peptides from vacuolar marker proteins, such as the vacuolar polyP polymerase subunit Vtc3 (35-fold), the vacuolar amino acid transporter Avt3 (48-fold), the v-ATPase subunit Vma9 (46-fold), or the alkaline phosphatase Pho8 (34-fold). By contrast, typical plasma membrane proteins were barely enriched, such as the iron permease Ftr1 (2.5-fold), the P$_i$ importer Pho87 (2.4-fold), or the polyamine importer Tpo5 (2.2-fold) (***Supplementary file 1***). Its co-enrichment with vacuolar proteins suggests that the major fraction of non-tagged Pho91 indeed resides in the vacuole. In agreement with this, Pho91 engages the AP3-dependent vesicular trafficking pathway, which leads from the Golgi to the vacuole (***Eising et al., 2022***). Thus, Pho91 is indeed a vacuolar P$_i$ transporter, but C- or N-terminal protein fusions easily interfere with correct sorting and cannot serve as reliable reporters for localisation of this protein.

We tested the impact of Pho91 on P$_i$ accumulation by vacuoles in vitro, using the same approach as above (***Figure 5***). Vacuoles lacking Pho91 (*pho91Δ*) accumulated P$_i$ two times faster than the wildtype and the maximal accumulated concentration was two times higher. This is consistent with a function of Pho91 as vacuolar P$_i$ exporter (***Hürlimann et al., 2007***; ***Potapenko et al., 2018***; ***Wang et al., 2015***). The difference between *pho91Δ* and wildtype vacuoles vanished when the vacuoles were incubated

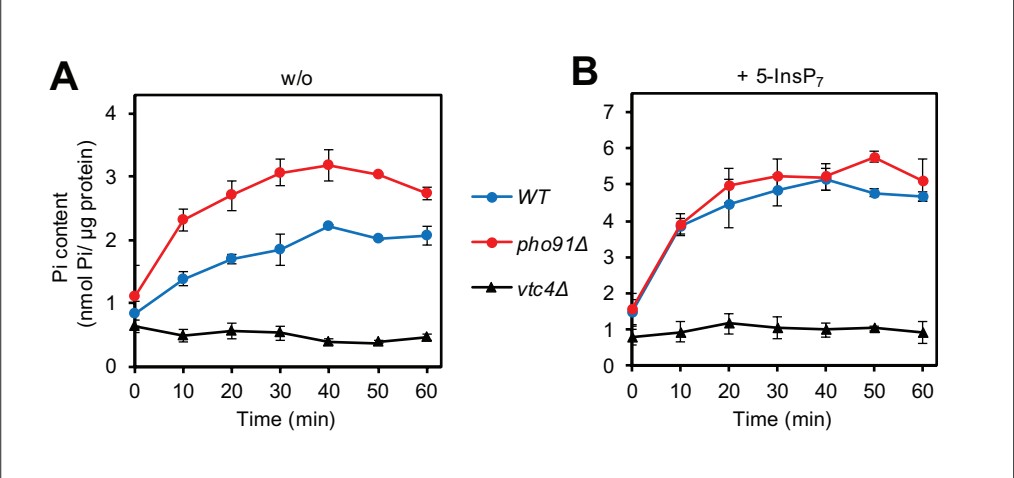

**Figure 5.** Accumulation of $P_i$ in isolated vacuoles. VTC and Pho91 dependence. Vacuoles were isolated from the indicated logarithmically growing strains. The purified organelles were incubated as in *Figure 1*, i.e., in a buffer with an ATP-regenerating system that allows the synthesis of polyphosphates (polyP), and either (**A**) without (w/o) or (**B**) in the presence of 50 µM 5-$IP_7$. After the indicated periods of incubation at 27°C, an 80 µl aliquot was withdrawn, the vacuoles were sedimented by centrifugation, washed and lysed. Released vacuolar $P_i$ was determined by malachite green assay. Graphs represent the mean ± SEM of three independent experiments. $p<0.05$ from a paired t-test comparing each time WT vs *pho91Δ*. (**B**) The indicated yeast strains were logarithmically grown overnight in synthetic complete (SC) medium with 7.5 mM $P_i$, harvested at $OD_{600nm}=1$, and immediately imaged by fluorescence microscopy as in A. The graph shows the means and SEM of the percentage of cells showing Pho4 predominantly in the nucleus. n=3 independent experiments with 200 cells quantified per sample. **$p<0.005$, ***$p<0.001$, ****$p<0.0001$ from a paired t-test comparing WT with the mutants.

in the presence of the inositol pyrophosphate 5-$InsP_7$. Since, as we showed above, vacuolar $P_i$ accumulation depends on polyP synthesis through VTC, the relative enhancement of $P_i$ accumulation in wildtype vacuoles through 5-$InsP_7$ could be caused by downregulation of Pho91, which would make the wildtype vacuoles behave similarly to *pho91Δ* vacuoles. Alternatively, 5-$InsP_7$ might stimulate polyP synthesis in wildtype more than in *pho91Δ* vacuoles. We could rule out the latter explanation based on two observations: An assay of polyP synthesis during the incubation (*Figure 6*) revealed that 5-$InsP_7$ stimulated polyP synthesis in wildtype and *pho91Δ* vacuoles to similar degrees. Furthermore, as shown above (*Figure 2*), polyP synthesis activity in wildtype cells is not rate-limiting for vacuolar

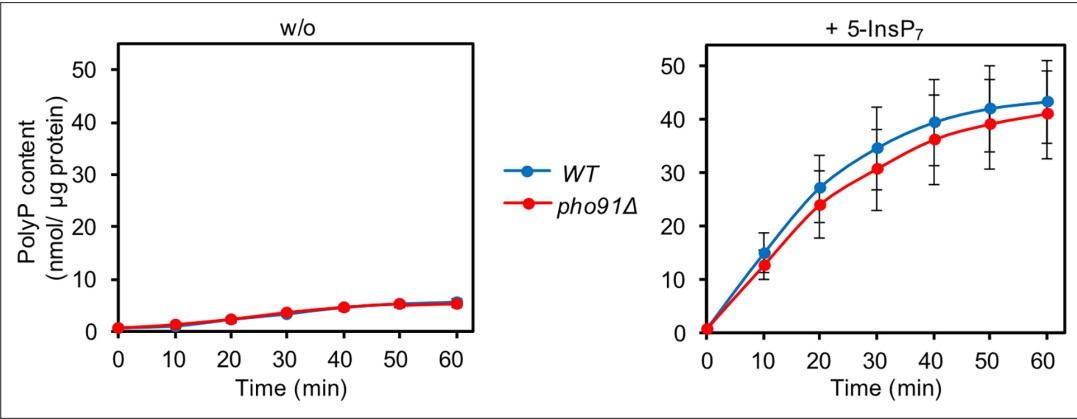

**Figure 6.** Polyphosphate (PolyP) accumulation in pho91 mutant vacuoles. Vacuoles from BY4742 cells (WT) and isogenic *pho91Δ* cells were isolated and incubated under conditions supporting polyP synthesis and $P_i$ accumulation as in *Figure 5*, in the absence (w/o) or presence of 50 µM 5-$InsP_7$. At the indicated time points, aliquots were withdrawn, the vacuoles were lysed in detergent, and polyP was assayed through DAPI fluorescence. Graphs show the means and SEM from three independent experiments. $p>0.05$ from a paired t-test for all differences between WT and *pho91Δ*.

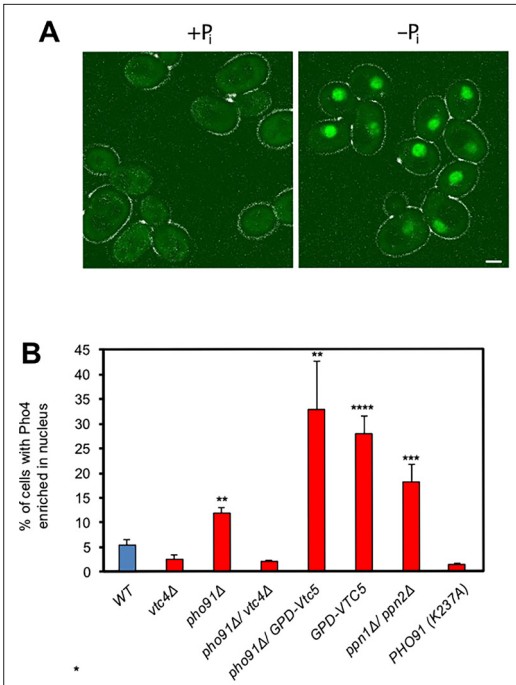

**Figure 7.** Impact of the vacuolar polyP/$P_i$ cycle on cytosolic $P_i$ signalling. (**A**) Illustration of nucleo-cytoplasmic relocation of Pho4-GFP in response to $P_i$ availability. Wildtype yeast cells were grown in synthetic complete (SC) medium under $P_i$-replete conditions. During the exponential phase ($OD_{600nm}$=1), cells were transferred for 30 min to SC media with 200 µM phosphate (-$P_i$) or 7.5 mM $P_i$ (+$P_i$) and imaged by fluorescence microscopy. Scale bar: 1 µm. (**B**) The indicated yeast strains were logarithmically grown over night in SC medium with 7.5 mM $P_i$, harvested at $OD_{600nm}$=1 and immediately imaged by fluorescence microscopy as in A. The graph shows the means and SEM of the percentage of cells showing Pho4 predominantly in the nucleus. n=3 independent experiments with 200 cells quantified per sample. ** $p<0.005$, *** $p<0.001$, ****$p<0.0001$ from a paired t-test comparing WT with the mutants.

$P_i$ accumulation. Therefore, we attribute the enhancement of $P_i$ accumulation through 5-InsP$_7$ to Pho91. Upon $P_i$ scarcity, the declining inositol pyrophosphate levels should then activate Pho91 to replenish cytosolic $P_i$.

## Interference with vacuolar polyP turnover provokes cytosolic $P_i$ scarcity

We tested the effects of this postulated cycle of polyP synthesis, polyphosphatase activity, and Pho91-mediated $P_i$ export on $P_i$ homeostasis in vivo. Pho4-GFP was used as a reporter, because this transcription factor shuttles between nucleus and cytosol. Under cellular $P_i$ scarcity and correspondingly low inositol pyrophosphate levels, it is predominantly nuclear, but it shifts to the cytosol under $P_i$-replete conditions (*Auesukaree et al., 2004*; *Chabert et al., 2023*; *Desfougères et al., 2016a*; *O'Neill et al., 1996*). Pho4-GFP localisation can hence serve as a readout for cytosolic $P_i$ signalling. In wildtype cultures growing logarithmically, cells can experience a transient shortage of $P_i$ during S-phase, when $P_i$ utilisation for biosynthesis may exceed the uptake capacity of the cell and consume its polyP stores (*Bru et al., 2016*; *Neef and Kladde, 2003*; *Pondugula et al., 2009*). Therefore, even on $P_i$-rich media, some cells can show nuclear accumulation of Pho4-GFP (*Vardi et al., 2014*; *Vardi et al., 2013*; *Wykoff et al., 2007*). Since S-phase occupies only around 20% of the yeast cycle (*Donaldson et al., 1998*; *Koren et al., 2010*), this fraction is expected to be correspondingly small. However, it is significant, and it can be clearly quantified by microscopy, which scores individual cells and not the population as an ensemble. We exploited this natural fluctuation of intracellular $P_i$ availability as a sensitising situation to assay how the vacuolar systems influence $P_i$ balance in the cytosol.

We scored the fraction of logarithmically growing cells that show nuclear Pho4-GFP even in $P_i$-replete medium. This fraction was 7% for wild-type cells (*Figure 7*). It increased fourfold upon hyperactivation of polyP synthesis, which we achieved by overexpression of the regulatory VTC subunit Vtc5 (*Desfougères et al., 2016a*). Ablation of polyP synthesis by deletion of the catalytic subunit Vtc4 had the opposite effect and reduced the frequency of cells with nuclear Pho4-GFP by half. Deletion of the vacuolar polyphosphatases Ppn1 and Ppn2 prevents polyP turnover and leads to the accumulation of extremely long polyP chains (*Gerasimaitė et al., 2017*; *Sethuraman et al., 2001*). We may thus expect phosphate to remain fixed in the form of polyP instead of being made available for $P_i$ reflux into the cytosol.

In line with this, the fraction of *ppn1Δ ppn2Δ* double mutants that showed nuclear Pho4-GFP was threefold higher than in wildtype. *pho91Δ* cells, in which we expect $P_i$ export from the vacuoles to be impaired, showed two times higher frequency of nuclear Pho4-GFP than wildtype. *pho91Δ vtc4Δ* double mutants showed an even lower frequency of nuclear Pho4-GFP than wildtype, suggesting that the state of $P_i$ starvation that *pho91Δ* favours is dependent on vacuolar polyP accumulation. Cells expressing *pho91$^{K237A}$* as the sole source of Pho91 also showed a 50% lower frequency of nuclear

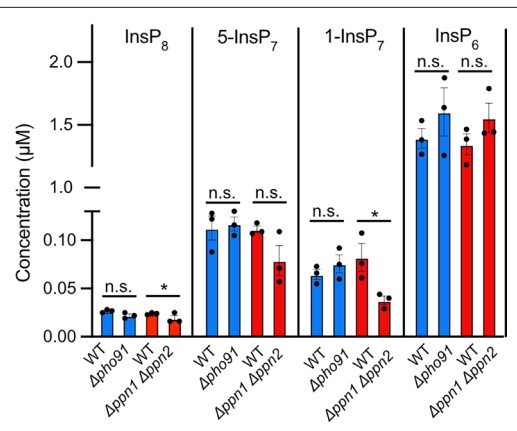

**Figure 8.** Impact of Pho91 and vacuolar polyphosphatases on InsPP levels. *pho91Δ* and *ppn1Δppn2Δ* cells, as well as their isogenic wildtypes, were logarithmically grown in $P_i$-replete synthetic complete (SC) medium as in **Figure 7**. At $OD_{600nm}=1$, the cells were extracted with perchloric acid as previously described (**Wilson et al., 2015**) and analysed for the indicated InsPs through capillary electrophoresis coupled to mass spectrometry (CE-MS) as described (**Qiu et al., 2023**). n=3. *p<0.05 from a paired t-test comparing WT with the mutants.

Pho4-GFP. The *pho91^K237A* allele generates an amino acid substitution in the InsPP-binding patch of the Pho91 SPX domain. It mimics the InsPP-free state (**Wild et al., 2016**) and hence low-$P_i$ conditions (**Chabert et al., 2023**). Collectively, our results are consistent with the notion that loss of InsPP binding activates Pho91 to export $P_i$ from vacuoles to the cytosol, enhancing repression of the PHO pathway.

Since InsPP levels in yeast decline in response to cellular $P_i$ availability (**Chabert et al., 2023**; **Lonetti et al., 2011**), we tested the impact of Pho91, Ppn1, and Ppn2 on the cellular levels of these metabolites using capillary electrophoresis coupled to mass spectrometry (CE-MS) (**Qiu et al., 2023**; **Qiu et al., 2021**; **Qiu et al., 2020**). The cells were harvested from the same $P_i$-replete growth conditions as for the microscopic assays above (**Figure 8**). The *ppn1Δ ppn2Δ* double mutants showed a decrease of 30–50% in the inositol pyrophosphates 1,5-InsP$_8$, 5-InsP$_7$, and 1-InsP$_7$, whereas InsP$_6$ remained at a similar level as in wildtype. InsPP changes at this scale are functionally significant, because they suffice to trigger the initial phase of the $P_i$ starvation response (**Chabert et al., 2023**; **Kim et al., 2025**; **Kim et al., 2023**). This is consistent with the threefold increase of cells with nuclear Pho4-GFP (**Figure 7**). *pho91Δ* cells did not show significant changes for all four metabolites although they showed a partial shift of Pho4-GFP into the nucleus (**Figures 7 and 8**). We attribute this discrepancy to the different nature of the assays. The microscopic assay for Pho4-GFP localisation can pick up effects in a fraction of the cells because it offers single-cell resolution. Inositol pyrophosphate analysis is an ensemble assay, in which changes in a smaller fraction of the population become diluted through the major pool that does not show the effect. For this reason, a change in inositol pyrophosphates affecting only 10% of the *pho91Δ* cells that show nuclear Pho4-GFP remains undetectable. The microscopic assay suggests that cytosolic $P_i$ scarcity in *ppn1Δ ppn2Δ* affects more cells, perhaps because it is more profound, and it hence becomes detectable even in a whole population analysis.

## Discussion

Our observations can be integrated with existing data on the properties of VTC (**Gerasimaitė et al., 2017**; **Gerasimaitė et al., 2014**; **Guan et al., 2023**; **Hothorn et al., 2009**; **Liu et al., 2023**; **Müller et al., 2002**; **Pipercevic et al., 2023**; **Wild et al., 2016**) and Pho91 (**Hürlimann et al., 2007**; **Potapenko et al., 2018**; **Wang et al., 2015**) to generate a working model explaining how an acidocalcisome-like organelle such as the yeast vacuole is set up to function as a $P_i$ buffer for the cytosol (**Figure 9**). Under $P_i$-replete conditions, high InsPP levels activate VTC to polymerise $P_i$ into polyP and translocate it into the vacuolar lumen. Here, the vacuolar polyphosphatases degrade polyP into $P_i$, filling the lumen with $P_i$ (**Gerasimaitė et al., 2017**; **Lichko et al., 2010**; **Sethuraman et al., 2001**; **Shi and Kornberg, 2005**). Since the $P_i$ exporter Pho91 is downregulated through InsPP binding to its SPX domain (**Hürlimann et al., 2007**; **Potapenko et al., 2018**; **Wang et al., 2015**), the $P_i$ liberated through polyP hydrolysis accumulates in the vacuoles. Product inhibition of the polyphosphatases attenuates polyP hydrolysis once the vacuolar lumen has reached a $P_i$ concentration above 30 mM. When the cells experience $P_i$ scarcity, InsPP levels decline (**Chabert et al., 2023**). This activates Pho91 to release $P_i$ from the vacuolar pool into the cytosol and stabilises cytosolic $P_i$.

Yeast cells do not only accumulate $P_i$ as a rapidly accessible buffer for the cytosol. Under $P_i$-replete conditions, they accumulate hundreds of millimolar of phosphate in the form of polyP (**Urech et al.,**

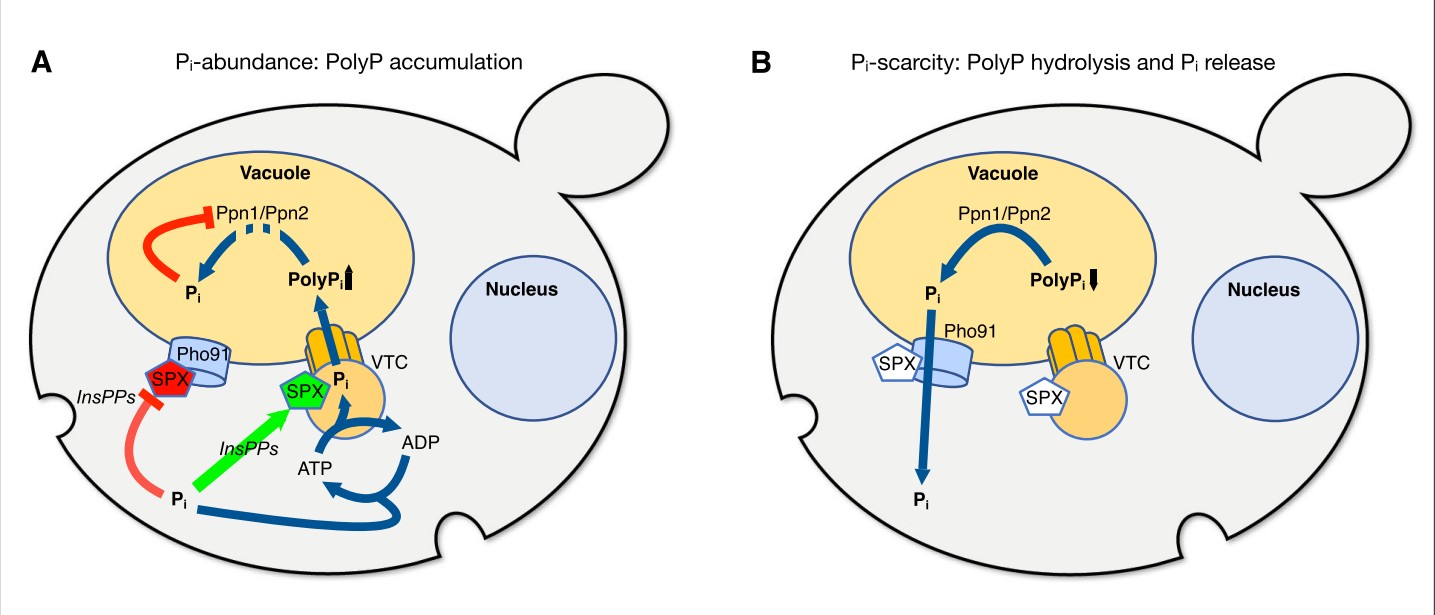

**Figure 9.** Working model of acidocalcisome-like vacuoles as $P_i$ buffering systems. (**A**) Under $P_i$-replete conditions, ATP drives the conversion of $P_i$ into polyphosphates (polyP) and its translocation into the organelle. Here, polyP is degraded by the vacuolar polyphosphatases Ppn1 and Ppn2 to establish a vacuolar pool of free $P_i$. Feedback inhibition of $P_i$ gradually reduces polyP degradation, enabling the buildup of a vacuolar polyP stock. Red lines and SPX colouring indicate inhibitory action, green colouring stimulation. (**B**) Cytosolic $P_i$ scarcity decreases InsPP levels, which triggers two compensatory, SPX-controlled effects: The transfer of $P_i$ from the cytosol into vacuoles through VTC ceases, and Pho91-dependent export of $P_i$ from vacuoles is activated. Both measures synergise to stabilise cytosolic $P_i$. The export of $P_i$ from the vacuole lifts product inhibition on the polyphosphatases Ppn1 and Ppn2 and stimulates a compensatory degradation of polyP.

1978). In contrast to the vacuolar reserve of $P_i$, which is presumed to be accessible immediately, mobilising the polyP store takes minutes to hours (*Bru et al., 2016*; *Nicolay et al., 1982*; *Pondugula et al., 2009*). But the polyP store offers advantages in the form of high capacity – hundreds of millimolar of phosphate units can be stored in the form of polyP – and low osmotic activity of polyP (*Dürr et al., 1979*). Keeping such a large stock of a critical resource, which is often growth-limiting in nature, is relevant for the cells. In case of phosphate shortage, the vacuolar polyP store can be mobilised to enable the cells to complete the cell cycle and transition into $G_0$ phase (*Müller et al., 1992*; *Jiménez et al., 2015*; *Westenberg et al., 1989*). This can consume substantial amounts of phosphate, because we can estimate that replicating the entire DNA ($1.2*10^7$ base pairs) immobilises roughly 1 mM $P_i$ in the cells, and cellular RNA is even 50 times more abundant than DNA (*Warner, 1999*), accounting for 50 mM phosphate. Phospholipids, which must also be synthesised to complete a cell cycle, fix phosphate in similar amounts (*Lange and Heijnen, 2001*). Thus, a large polyP store is necessary to guarantee that the cells can finish S-phase upon a shortage of phosphate sources. In accordance with this notion, the absence of the polyP store impairs cell cycle progression and nucleotide synthesis, and induces genome instability (*Bru et al., 2017*; *Bru et al., 2016*). Also, a shift from non-fermentable carbon sources to fermentation of glucose leads to a strong requirement for $P_i$ because the activation of glucose uptake and glycolysis depends on large amounts of phosphate-containing sugars and glycolytic intermediates (*Gillies et al., 1981*; *Nicolay et al., 1983*; *Nicolay et al., 1982*). A shortage of $P_i$ restrains the abundance of these metabolites (*Kim et al., 2023*).

The properties of the regulatory circuit described above imply an inbuilt switch from vacuolar $P_i$ accumulation to large-scale stocking of vacuolar polyP. $P_i$-replete conditions generate high cellular InsPP levels. These will not only reduce $P_i$ efflux from the vacuoles through Pho91 and inactivate the vacuolar polyphosphatases, but at the same time stimulate continued polyP synthesis by VTC. Coincidence of these effects will favour storage and high accumulation of phosphate in the form of polyP. Conversely, depletion of the vacuolar $P_i$ reservoir upon $P_i$ scarcity in the medium will activate the vacuolar polyphosphatases. In combination with the downregulation of the polyP polymerase VTC

through the decline of InsPPs, this will mobilise the large vacuolar polyP reserve once the immediately available vacuolar $P_i$ pool is gradually depleted.

The concentration of $P_i$ inside vacuoles as a rapidly accessible $P_i$ reserve, and the synthesis of a large polyP stock, comes at an energetic cost because the transformation of $P_i$ into polyP requires the formation of phosphoric anhydride bonds (*Gerasimaitė et al., 2014*; *Hothorn et al., 2009*) and vacuolar $P_i$ reaches 30 mM. This exceeds the cytosolic $P_i$ concentration, which was measured through $^{31}$P-NMR in a variety of yeasts, yielding values of 5–17 mM (*Nicolay et al., 1983*; *Nicolay et al., 1982*). Cytosolic $P_i$ can also be estimated based on data from several other studies (*Auesukaree et al., 2004*; *Hürlimann et al., 2009*; *Pinson et al., 2004*; *Theobald et al., 1996*; *Zhang et al., 2015*). Assuming that the cytosolic volume of a BY4741 yeast cell is 40 fL (*Chabert et al., 2023*), and 1 g of dry weight contains 40*10$^9$ yeast cells, these studies point to cytosolic values of 10–15 mM in $P_i$-replete media. Upon $P_i$ starvation, this value rapidly drops up to fivefold, resulting in a strong $P_i$ gradient across the vacuolar membrane (*Okorokov et al., 1980*; *Shirahama et al., 1996*). To replenish the cytosolic pool under $P_i$ scarcity, Pho91 can exploit not only this $P_i$ concentration gradient, but also the vacuolar electrochemical potential, which was shown to stimulate $P_i$ export through the Pho91 homologue OsSPX-MFS3 from plant vacuoles (*Wang et al., 2015*).

Vacuolar $P_i$ accumulation is driven indirectly through ATP in two ways. VTC uses ATP as a substrate and transfers the phosphoric anhydride bond of the γ-phosphate onto a polyP chain (*Hothorn et al., 2009*). The growing polyP chain exits from the catalytic site directly towards the transmembrane part of VTC (*Guan et al., 2023*; *Liu et al., 2023*). This transmembrane part likely forms a controlled channel that can guide polyP through the membrane (*Liu et al., 2023*). Coupled synthesis and translocation require the V-ATPase (*Gerasimaitė et al., 2014*), probably because polyP is highly negatively charged and therefore follows the electrochemical potential across the vacuolar membrane of 180 mV (inside positive) and 1.7 pH units (*Kakinuma et al., 1981*), which is generated through the proton pumping V-ATPase. Thus, the combination of the VTC complex and vacuolar polyphosphatases can be considered as a $P_i$ pump that is driven by ATP through polyP synthesis and through the electrochemical potential for polyP translocation and $P_i$ export.

It is likely that acidocalcisome- and lysosome-like organelles of other organisms act as buffers for cytosolic $P_i$ similarly as described in our model for yeasts. This notion is supported by the conserved molecular setup of acidocalcisome-like organelles, as well as by phenotypic similarities. The acidocalcisomes of trypanosomes contain VTC, a Pho91 homologue and proton pumps in their membranes, and polyphosphatases in their lumen (*Billington et al., 2023*; *Fang et al., 2007*; *Huang and Docampo, 2015*; *Lander et al., 2013*; *Scott et al., 1997*; *Ulrich et al., 2013*). Also, the acidocalcisome-like organelles of the alga *Chlamydomonas* contain such proteins and they accumulate polyP through VTC as a function of the availability of $P_i$, a proton gradient, and metal ions (*Aksoy et al., 2014*; *Blaby-Haas and Merchant, 2014*; *Goodenough et al., 2019*; *Hong-Hermesdorf et al., 2014*; *Long et al., 2023*; *Ruiz et al., 2001*; *Zúñiga-Burgos et al., 2024*). Like in yeast, the polyP stores are mobilised upon $P_i$ limitation (*Plouviez et al., 2021*; *Sanz-Luque et al., 2020*). *Drosophila* has a potentially lysosome-related compartment, which is acidic, carries V-ATPase and a homologue of the Pi exporter XPR1, impacts cytosolic $P_i$, and diminishes upon $P_i$ starvation (*Xu et al., 2023*). Mammalian lysosome-like organelles also participate in $P_i$ homeostasis. They can accumulate polyP and take up $P_i$ (*Pisoni, 1991*; *Pisoni and Lindley, 1992*). They carry the $P_i$ exporter XPR1, which interacts with the plasma membrane $P_i$ importer PiT1 to regulate its degradation (*Li et al., 2024*).

We hence propose that acidocalcisome-like vacuoles may have a general role as feedback-controlled, rapidly accessible $P_i$ buffers for the cytosol, addressing a critical parameter for metabolism. However, given that acidocalcisome-like organelles accumulate not only phosphate but also multiple other metabolites and ions (*Docampo, 2024*), they are probably interlinked with cellular metabolism in multiple ways and might form an important hub for its homeostasis.

## Materials and methods
### Materials and data availability

All strains used in this study (listed in *Supplementary file 1*) are available from the corresponding author upon request. Source data from the proteomic analysis have been deposited at the PRIDE database under the identifier PXD060102.

**Table 1.** Yeast strains used in this study.

| Background | Genotype | Source |
|---|---|---|
| BY4741 | MATa his3Δ1 leu2Δ0 met15Δ0 ura3Δ0 | Euroscarf |
| BY4742 | MATα his3Δ1 leu2Δ0 lys2Δ0 ura3Δ0 | Euroscarf |
| BY4742 | *vtc4::kanMX* | *Gerasimaitė et al., 2017* |
| BY4741 | *vtc4::kanMX* | This study |
| BY4742 | *ppn1::natNT2* | *Gerasimaitė et al., 2017* |
| BY4742 | *ppn2::URA3* | *Gerasimaitė et al., 2017* |
| BY4742 | *ppn1::natNT2 ppn2::URA3* | *Gerasimaitė et al., 2017* |
| BY4742 | *ppn1::natNT2 ppn2::URA3 vtc4::URA3* | *Gerasimaitė et al., 2017* |
| BY4741 | P$_{GPD1}$.VTC5 natNT2 | *Desfougères et al., 2016a* |
| BY4742 | VTC3$^{K126A}$ VTC4$^{K129A}$ | *Wild et al., 2016* |
| BY4741 | *pho91::kanMX* | This study |
| BY4741 | *pho91::kanMX* P$_{GPD1}$.VTC5 natNT2 | This study |
| BY4741 | *pho91::kanMX* pRS303-P$_{PHO91}$-PHO91 (K237A) | This study |
| BY4741 | *pho91::hyg vtc4::kanMX* | This study |
| BY4741 | pRS415-P$_{PHO4}$-PHO4-GFP | This study |
| BY4741 | *vtc4::kanMX* pRS415-P$_{PHO4}$-PHO4-GFP | This study |
| BY4741 | *pho91::kanMX* pRS415-P$_{PHO4}$-PHO4-GFP | This study |
| BY4741 | *pho91::hyg vtc4::kanMX* pRS415-P$_{PHO4}$-PHO4-GFP | This study |
| BY4741 | P$_{GPD1}$-VTC5 natNT2 *pho91::hyg* pRS415-P$_{PHO4}$-PHO4-GFP | This study |
| BY4741 | P$_{GPD1}$-VTC5 natNT2 pRS415-P$_{PHO4}$-PHO4-GFP | This study |
| BY4742 | *ppn1::natNT2 ppn2::URA3* pRS415-P$_{PHO4}$-PHO4-GFP | This study |
| BY4741 | *pho91::kanMX* pRS303-P$_{PHO91}$-PHO91 (K237A) pRS415-P$_{PHO4}$-PHO4-GFP | This study |

## Strains

The *Saccharomyces cerevisiae* strains used in this study (*Table 1*) were obtained by genetic manipulations from the BY4741 and BY4742 background, which represent the 'wildtype' of this study. For the sake of readability, the background was not indicated in the genotype of each mutant strain. Genetic manipulations of yeast were performed by homologous recombination according to published procedures and/or transformation with the indicated plasmids (*Gietz and Schiestl, 2007*; *Güldener et al., 1996*).

## Growth conditions

*S. cerevisiae* cells were grown on yeast extract-peptone-dextrose (YPD: 1% yeast extract, 2% peptone, and 2% dextrose) or in synthetic complete (SC) medium from.

## PHO4 localisation

Cells transformed with pRS415-P$_{PHO4}$-PHO4-GFP, a plasmid expressing a PHO4-GFP fusion from the PHO4 promoter (*Chabert et al., 2023*), were grown exponentially overnight in SC medium without leucine (SC-Leu). Care was taken that the culture did not grow beyond a density of OD$_{600}$=0.7. At this point, the cells were collected by brief centrifugation (15 s, 3000 × *g*) in a tabletop centrifuge and resuspended in 1/10th to 1/20th of their own supernatant. Pho4-GFP localisation was immediately checked by microscopy. An aliquot of the cells was washed twice with SC-Leu without P$_i$ and then diluted in SC-Leu with 200 µM phosphate to OD$_{600}$=0.7. After 30 min of incubation in this medium,

Pho4-GFP localisation was analysed by fluorescence microscopy on a LEICA DMI6000B inverted microscope equipped with a Hamamatsu ORCA-R2 (C10600-10B) camera, an XCite series 120Q UV lamp, and a Leica 100× 1.4 NA lens.

## Vacuole preparation

Vacuoles were purified from yeast cells essentially as described (*D'Agostino and Mayer, 2019*). Briefly, yeast cells were grown in 1 l of YPD to an $OD_{600}$ of 1.5. 330 ml of cells were collected by centrifugation and resuspended in 50 ml of 30 mM Tris-HCl pH 8.9, 10 mM DTT buffer. Suspensions were incubated in a 30°C water bath for 5 min and then collected by centrifugation. The pellet was resuspended in 15 ml of spheroplasting solution (50 mM K-phosphate pH 7.5, 600 mM sorbitol in YPD with 0.2% D-glucose and 3600 U/ml lyticase) and incubated for 25 min at 30°C. Spheroplasts were collected by centrifugation (2500 × $g$, 3 min) and resuspended in 2.5 ml of 15% Ficoll 400 in PS buffer (10 mM PIPES-KOH pH 6.8, 200 mM sorbitol). 80 µg of DEAE-dextran were added under gentle mixing. After incubation on ice for 2 min and then at 30°C for 80 s, spheroplasts were transferred into Beckman SW41.1 tubes, overlaid with cushions of 8%, 4%, and 0% Ficoll 400 in PS buffer, and centrifuged (150,000 × $g$, 90 min, 2°C). Vacuoles were collected from the 0–4% Ficoll interface. Their protein concentration was determined through Bradford assay using BSA as a standard.

## Polyphosphatase activity of vacuolar lysates

Polyphosphatase activity was assayed as described previously (*Gerasimaitė et al., 2017*), with the following modifications. Isolated vacuoles were diluted to a final protein concentration of 0.002 mg/ml in 1 ml of reaction buffer (20 mM PIPES/KOH pH 6.8, 150 mM KCl, 1 mM $ZnCl_2$ or $MgCl_2$, 0.1% Triton X-100, 1xPIC, 1 mM PMSF, and 30 or 300 µM $polyP_{300}$) in the presence or absence of various concentrations of $KH_2PO_4$ and incubated at 27°C. At the indicated times, 80 µl aliquots were collected and the reaction was stopped by dilution with 160 µl of stop solution (10 mM PIPES/KOH pH 6.8, 150 mM KCl, 12 mM EDTA pH 8.0, 0.1% Triton X-100, 15 µM DAPI). Remaining polyP was quantified by measuring polyP-DAPI fluorescence ($\lambda_{exc.}$ 415 nm, $\lambda_{em.}$ 550 nm) in a black 96-well plate in a Spectramax Gemini microplate fluorometer (Molecular Devices). A reaction containing boiled vacuoles was used as a negative control.

## Polyphosphate synthesis by isolated vacuoles

Polyphosphate synthesis was assayed as described (*Gerasimaitė et al., 2014*). Isolated vacuoles were diluted to final protein concentration of 0.02 mg/ml on 1 ml of reaction buffer (10 mM PIPES/KOH pH 6.8, 150 mM KCl, 0.5 mM $MnCl_2$, 200 mM sorbitol) and the reaction was started by adding an ATP regenerating system (1 mM ATP-$MgCl_2$, 40 mM creatine phosphate, and 0.25 mg/ml creatine kinase). The mix was incubated at 27°C. At different time points, 80 µl aliquots were mixed with 160 µl of stop solution (10 mM PIPES/KOH pH 6.8, 150 mM KCl, 200 mM sorbitol, 12 mM EDTA, 0.15% Triton X-100, and 15 µM DAPI). PolyP synthesis was quantified through polyP-DAPI fluorescence ($\lambda_{exc.}$ 415 nm, $\lambda_{em.}$ 550 nm) in a black 96-well plate. A calibration curve was prepared using commercial $polyP_{60}$ as a standard.

## Phosphate quantification in isolated vacuoles

Isolated vacuoles were incubated as described for the polyP synthesis assay above. At different time points, 80 µl aliquots were centrifuged (3 min, 2000 × $g$, 2°C), the pellets were washed with 500 µl of washing solution (10 mM PIPES/KOH pH 6.8, 200 mM sorbitol, 150 mM KCl) and centrifuged as before. The final pellet was resuspended with 100 µl of lysis buffer (10 mM PIPES/KOH pH 6.8, 200 mM sorbitol, 150 mM KCl, 12 mM EDTA, 0.1% Triton). Free phosphate was quantified by adding 150 µl of molybdate-malachite green solution (1 mM malachite green, 10 mM ammonium molybdate, 1M HCl) and reading the absorbance at 595 nm in a microplate photometer.

## Inositol pyrophosphate synthesis, extraction, and quantification

5-InsP7 was synthesised as described (*Capolicchio et al., 2013*; *Wang et al., 2014*). For quantification from cells, InsPPs extraction was performed as described previously (*Kim et al., 2023*), with the following modifications. Briefly, 3 ml of yeast cell culture at $OD_{600nm}=1$ was collected using rapid vacuum-filtration on a polytetrafluoroethylene membrane filter (1.2 µm; Piper Filter GmbH, Germany).

After snap-freezing in liquid nitrogen, yeast cells on the membrane were resuspended in 400 µl of 1 M perchloric acid and lysed by bead beating (glass beads; 0.25–0.5 mm) for 10 min at 4°C. After centrifugation at 13,000 rpm for 3 min at 4°C, the supernatant was transferred into a new tube containing 3 mg of titanium dioxide ($TiO_2$) beads (GL Sciences, Japan) which had been pre-washed twice with $H_2O$ and 1 M perchloric acid. The sample was gently rotated for 15 min at 4°C. The $TiO_2$ beads were collected by centrifugation at 13,000 rpm for 1 min at 4°C and washed twice using 1 M perchloric acid. After the second washing step, the $TiO_2$ beads were resuspended in 300 µl of 3% (vol/vol) $NH_4OH$ and rotated gently at room temperature. After centrifugation at 13,000 rpm for 1 min, the eluants were transferred into a new tube and dried in SpeedVac (Labogene, Denmark) at 42°C. InsPPs were measured through CE-MS as described (*Qiu et al., 2021*).

## Quantification of the vacuolar proteome

Vacuoles were prepared as described above. During the purification procedure, a sample of the sedimented spheroplasts was withdrawn before DEAE-dextran was added. These withdrawn spheroplasts constitute the 'whole-cell' extract. After the flotation step, the vacuoles withdrawn from the 4%–0% Ficoll interface were used as the vacuole fraction. For each fraction sample, 100 µg protein was precipitated by adding a final concentration of 12.5% TCA for 10 min on ice. The proteins were sedimented (12,000 × *g*, 5 min, room temperature), the supernatant discarded, and the pellets were washed with ice-cold acetone and centrifuged as above twice. The final pellet was dried and dissolved in reducing SDS sample buffer (10% glycerol, 50 mM Tris-HCl pH 6.8, 2 mM DTT, 2% SDS, 0.002% bromophenol blue).

Samples dissolved in Laemmli buffer (50 mM Tris pH 6.8, 10 mM DTT, 2% SDS, 0.1% bromophenol blue, 10% glycerol) were equalised in concentration by dilution in SP3 buffer (2% SDS, 10 mM DTT, 50 mM Tris, pH 7.5). 150 µg of protein per sample were heated at 95°C for 5 min and cooled down. Reduced cysteine residues were alkylated by adding iodoacetamide (30 mM final) and incubating for 45 min at room temperature in the dark. Digestion was done by the SP3 method (*Hughes et al., 2019*) using magnetic Sera-Mag Speedbeads (Cytiva 45152105050250, 50 mg/ml). Beads were added at a ratio of 10:1 (wt:wt) to samples, and proteins were precipitated on beads with ethanol (final concentration: 60%). After three washes with 80% ethanol, beads were digested in 50 µl of 100 mM ammonium bicarbonate with 3.0 µg of trypsin (Promega #V5073). After 1 hr of incubation at 37°C, the same amount of trypsin was added to the samples for an additional 1 hr of incubation. Supernatants were then recovered, transferred to new tubes, acidified with formic acid (0.5% final concentration), and dried by centrifugal evaporation. To remove traces of SDS, two sample volumes of isopropanol containing 1% TFA were added to the digests, and the samples were desalted on a strong cation exchange plate (Oasis MCX; Waters Corp., Milford, MA, USA) by centrifugation. After washing with isopropanol/1% TFA and 2% acetonitrile/0.1% FA, peptides were eluted in 200 µl of 80% MeCN, 19% water, 1% (vol/vol) ammonia, and dried by centrifugal evaporation. Data-dependent LC-MS/MS analyses of samples were carried out on a Fusion Tribrid Orbitrap mass spectrometer (Thermo Fisher Scientific) interfaced through a nano-electrospray ion source to an Ultimate 3000 RSLCnano HPLC system (Dionex). Peptides were separated on a reversed-phase custom-packed 45 cm C18 column (75 µm ID, 100 Å, Reprosil Pur 1.9 µm particles, Dr. Maisch, Germany) with a 4–90% acetonitrile gradient in 0.1% formic acid at a flow rate of 250 nl/min (total time 140 min). Full MS survey scans were performed at 120,000 resolution. A data-dependent acquisition method controlled by Xcalibur software (Thermo Fisher Scientific) was used that optimised the number of precursors selected ('top speed') of charge 2+ to 5+ while maintaining a fixed scan cycle of 0.6 s. Peptides were fragmented by higher-energy collision dissociation (HCD) with a normalised energy of 32%. The precursor isolation window used was 1.6 Th, and the MS2 scans were done in the ion trap. The m/z of fragmented precursors was then dynamically excluded from selection during 60 s.

Data files were analysed with MaxQuant 1.6.14.0 (*Cox and Mann, 2008*) incorporating the Andromeda search engine (*Cox et al., 2011*). Cysteine carbamidomethylation was selected as fixed modification while methionine oxidation and protein N-terminal acetylation were specified as variable modifications. The sequence databases used for searching were the *S. cerevisiae* reference proteome based on the UniProt database (https://www.uniprot.org/, version of June 6, 2021, containing 6050 sequences), and a 'contaminant' database containing the most usual environmental contaminants and enzymes used for digestion (keratins, trypsin, etc.). Mass tolerance was 4.5 ppm on precursors (after

recalibration) and 20 ppm on MS/MS fragments. Both peptide and protein identifications were filtered at 1% FDR relative to hits against a decoy database built by reversing protein sequences. The match between runs feature was enabled.

## Acknowledgements

Mass spectrometry-based proteomics work was performed by the Protein Analysis Facility of the Faculty of Biology and Medicine, University of Lausanne. This study was supported by grants from the SNSF (320030-228119, 31003A_179306, and 310030_204713) and ERC (788442) to AM, by the HFSP (LT000588/2019) to GDK, by the Deutsche Forschungsgemeinschaft (CIBSS, EXC-2189, Project ID 390939984, to HJJ) and the Volkswagen Foundation (VW Momentum Grant 98604 to HJJ).

## Additional information

### Funding

| Funder | Grant reference number | Author |
| --- | --- | --- |
| Schweizerischer Nationalfonds zur Förderung der Wissenschaftlichen Forschung | 320030-228119 | Andreas Mayer |
| Schweizerischer Nationalfonds zur Förderung der Wissenschaftlichen Forschung | 31003A_179306 | Andreas Mayer |
| Schweizerischer Nationalfonds zur Förderung der Wissenschaftlichen Forschung | 310030_204713 | Andreas Mayer |
| European Research Council | 788442 | Andreas Mayer |
| Human Frontier Science Program | LT000588/2019 | Geun-Don Kim |
| Deutsche Forschungsgemeinschaft | 390939984 | Henning J Jessen |
| Volkswagen Foundation | Momentum Grant 98604 | Henning J Jessen |

The funders had no role in study design, data collection and interpretation, or the decision to submit the work for publication.

### Author contributions

Samuel Bru, Conceptualization, Formal analysis, Investigation, Methodology, Writing – original draft, Writing – review and editing; Lydie Michaillat Mayer, Data curation, Formal analysis, Validation, Investigation, Methodology, Writing – original draft; Geun-Don Kim, Formal analysis, Investigation, Methodology, Writing – original draft; Danye Qiu, Data curation, Formal analysis, Investigation, Methodology, Writing – original draft, Writing – review and editing; Henning J Jessen, Resources, Formal analysis, Supervision, Funding acquisition, Investigation, Methodology, Writing – original draft, Project administration, Writing – review and editing; Andreas Mayer, Resources, Data curation, Formal analysis, Supervision, Funding acquisition, Investigation, Methodology, Writing – original draft, Project administration, Writing – review and editing

### Author ORCIDs

Samuel Bru ID https://orcid.org/0000-0002-7005-8609
Lydie Michaillat Mayer ID https://orcid.org/0009-0008-3292-5899
Geun-Don Kim ID https://orcid.org/0000-0002-3580-2471

Henning J Jessen ⬚ https://orcid.org/0000-0002-1025-9484
Andreas Mayer ⬚ https://orcid.org/0000-0001-6131-313X

Reviewer #1 (Public review): https://doi.org/10.7554/eLife.108181.3.sa1
Reviewer #3 (Public review): https://doi.org/10.7554/eLife.108181.3.sa2
Author response https://doi.org/10.7554/eLife.108181.3.sa3

## Additional files

### Supplementary files
MDAR checklist

Supplementary file 1. Enrichment factors of marker proteins in the vacuolar fraction of BY4742 cells over the total extract of spheroblasts.

### Data availability
Data from the proteomic analysis have been deposited at the PRIDE database under the identifier PXD060102.

The following dataset was generated:

| Author(s) | Year | Dataset title | Dataset URL | Database and Identifier |
|---|---|---|---|---|
| Mayer LM | 2025 | Enrichment of proteins in purified *Saccharomyces cerevisiae* vacuoles relative to a whole cell extract | https://www.ebi.ac.uk/pride/archive/projects/PXD060102 | PRIDE, PXD060102 |

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
