## [Editor Report · eLife Assessment]

This study provides **fundamental** insights into eukaryotic phosphate homeostasis by demonstrating how yeast vacuoles dynamically regulate cytosolic phosphate levels. The conclusions are **convincing**, supported by an elegant combination of in vitro assays and in vivo measurements. This study will be of interest to cell biologists, particularly for those who are working in the field of phosphate metabolism.

---

## [Referee Report · Reviewer #1 (Public review)]

The manuscript by Bru et al. focuses on the role of vacuoles as a phosphate buffering system for yeast cells. The authors describe here the crosstalk between the vacuole and the cytosol using a combination of in vitro analyses of vacuoles and in vivo assays. They show that the luminal polyphosphatases of the vacuole can hydrolyze polyphosphates to generate inorganic phosphate, yet they are inhibited by high concentrations. This balances the synthesis of polyphosphates against the inorganic phosphate pool. Their data further show that the Pho91 transporter provides a valve for the cytosol as it gets activated by a decline in inositol pyrophosphate levels. The authors thus demonstrate how the vacuole functions as a phosphate buffering system to maintain a constant cytosolic inorganic phosphate pool.

This is a very consistent and well-written manuscript with a number of convincing experiments, where the authors use isolated vacuoles and cellular read-out systems to demonstrate the interplay of polyphosphate synthesis, hydrolysis, and release. The beauty of this system the authors present is the clear correlation between product inhibition and the role of Pho91 as a valve to release Pi to the cytosol to replenish the cytosolic pool. I find the paper overall an excellent fit.

Comments on Revision:

The authors have addressed all my concerns.

---

## [Referee Report · Reviewer #3 (Public review)]

Bru et al. investigated how inorganic phosphate (Pi) is buffered in cells using *S. cerevisiae* as a model. Pi is stored in cells in the form of polyphosphates in acidocalcisomes. In *S. cerevisiae*, the vacuole, which is the yeast lysosome, also fulfills the function of Pi storage organelle. Therefore, yeast is an ideal system to study Pi storage and mobilization.

They can recapitulate in their previously established system, using isolated yeast vacuoles, findings from their own and other groups. They integrate the available data and propose a working model of feedback loops to control the level of Pi on the cellular level.

This is a solid study, in which the biological significance of their findings is not entirely clear. The data analysis and statistical significance need to be improved and included, respectively. The manuscript would have benefited from rigorously testing the model, which would also have increased the impact of the study.

---

## [Author Response]

The following is the authors’ response to the original reviews.

**Reviewer #1 (Public review):**
The manuscript by Bru et al. focuses on the role of vacuoles as a phosphate buffering system for yeast cells. The authors describe here the crosstalk between the vacuole and the cytosol using a combination of in vitro analyses of vacuoles and in vivo assays. They show that the luminal polyphosphatases of the vacuole can hydrolyse polyphosphates to generate inorganic phosphate, yet they are inhibited by highconcentrations. This balances the synthesis of polyphosphates against the inorganic phosphate pool. Their data further show that the Pho91 transporter provides a valve for the cytosol as it gets activated by a decline in inositol pyrophosphate levels. The authors thus demonstrate how the vacuole functions as a phosphate buffering system to maintain a constant cytosolic inorganic phosphate pool.This is a very consistent and well-written manuscript with a number of convincing experiments, where the authors use isolated vacuoles and cellular read-out systems to demonstrate the interplay of polyphosphate synthesis, hydrolysis, and release. The beauty of this system the authors present is the clear correlation between product inhibition and the role of Pho91 as a valve to release Pi to the cytosol to replenish the cytosolic pool. I find the paper overall an excellent fit and only have a few issues, including:(1) Figure 3: The authors use in their assays 1 mM ZnCl2 or 1mM MgCl2. Is this concentration in the range of the vacuolar luminal ion concentration? Did they also test the effect of Ca2+, as this ion is also highly concentrated in the lumen?

The concentrations inside vacuoles reach those values. However, given that polyP can chelate divalent metal ions, what would matter are the concentrations of free Zn^2+^ or Mg^2+^ inside the organelle. These are not known. This is not critical since we use those two conditions only as a convenient tool to differentiate Ppn1 and Ppn2 activity in vitro. In our initial characterisation of Ppn2 (10.1242/jcs.201061), we had also tested Mn, Co, Ca, Ni, Cu. Only Zn and Co supported activity. Ca did not. Andreeva et al. (10.1016/j.biochi.2019.06.001) reached similar conclusions and extended our results.

(2) Regarding the concentration of 30 mM K-PI, did the authors also use higher and lower concentrations? I agree that there is inhibition by 30 mM, but they cannot derive conclusions on the luminal concentration if they use just one in their assay. A titration is necessary here.

The concentration of 30 mM was not chosen arbitrarily. It is the luminal P^i^ concentration that the vacuoles reached through polyP synthesis and hydrolysis when they entered a plateau of luminal P^i^. We consider this as an upper limit because polyP kept increasing which luminal P^i^ did not. Thus, there is no physiological motivation for trying higher values. We have nevertheless added a titration to the revised version (new Fig. 3A).

(3) What are the consequences on vacuole morphology if the cells lack Pho91?

We had not observed significant abnormalities during a screen of the genome-wide deletion collection of yeast (10.1371/journal.pone.0054160), nor in other experiments with pho91 mutants, which we have not included in this manuscript due to a lack of effect.

(4) Discussion: The authors do not refer to the effect of calcium, even though I would expect that the levels of the counterion should affect the phosphate metabolism. I would appreciate it if they would extend their discussion accordingly.

The situation is much more complex because Ca2+ is not the only counterion. Major pools of counterions (up to hundreds of mM) are constituted by vacuolar lysine, arginine, polyamines, Mg, Zn etc. Their interplay with polyP is probably complex and worth to be treated in a dedicated project. If we wanted to limit the discussion of this complexity not to the simple statement that it is not understood, which is not very useful, we would have to engage in a lot of speculation. We feel that this would make the discussion lose focus and not contribute concrete insights.

(5) I would appreciate a brief discussion on how phosphate sensing and control are done in human cells. Do they use a similar lysosomal buffer system?

Mammalian cells have their Pi exporter XPR1 mainly on a lysosome-like compartment (10.1016/j.celrep.2024.114316). Whether and how it functions there for Pi export from the cytosol is not entirely clear. We have addressed this situation in the revised discussion section.

**Reviewer #2 (Public review):**
Summary:This manuscript presents a well-conceived and concise study that significantly advances our understanding of polyphosphate (polyP) metabolism and its role in cytosolic phosphate (Pi) homeostasis in a model unicellular eukaryote. The authors provide evidence that yeast vacuoles function as dynamic regulatory buffers for Pi homeostasis, integrating polyP synthesis, storage, and hydrolysis in response to cellular metabolic demands. The work is methodologically sound and offers valuable insights into the conserved mechanisms of phosphate regulation across eukaryotes.Strengths:The results demonstrate that the vacuolar transporter chaperone (VTC) complex, in conjunction with luminal polyphosphatases (Ppn1/Ppn2) and the Pi exporter Pho91, establishes a finely tuned feedback system that balances cytosolic Pi levels. Under Pi-replete conditions, inositol pyrophosphates (InsPPs) promote polyP synthesis and storage while inhibiting polyP hydrolysis, leading to vacuolar Pi accumulation.Conversely, Pi scarcity triggers InsPP depletion, activating Pho91-mediated Pi export and polyP mobilization to sustain cytosolic phosphate levels. This regulatory circuit ensures metabolic flexibility, particularly during critical processes such as glycolysis, nucleotide synthesis, and cell cycle progression, where phosphate demand fluctuates dramatically.From my viewpoint, one of the most important findings is the demonstration that vacuoles act as a rapidly accessible Pi reservoir, capable of switching between storage (as polyP) and release (as free Pi) in response to metabolic cues. The energetic cost of polyP synthesis-driven by ATP and the vacuolar proton gradient-highlights the evolutionary importance of this buffering system. The study also draws parallels between yeast vacuoles and acidocalcisomes in other eukaryotes, such as Trypanosoma and Chlamydomonas, suggesting a conserved role for these organelles in phosphate homeostasis.Weaknesses:While the manuscript is highly insightful, referring to yeast vacuoles as "acidocalcisome-like" may warrant further discussion. Canonical acidocalcisomes are structurally and chemically distinct (e.g., electrondense, in most cases spherical, and not routinely subjected to morphological changes, and enriched with specific ions), whereas yeast vacuoles have well-established roles beyond phosphate storage. A comment on this terminology could strengthen the comparative analysis and avoid potential confusion in the field.

Yeast vacuoles show all major chemical features of acidocalcisomes. They are acidified, contain high concentrations of Ca, polyP (which make them electron-dense, too), other divalent ions, such as Mg, Zn, Mn etc, and high concentrations of basic amino acids. Thus, they clearly have an acidocalcisome-like character. In addition, they have hydrolytic, lysosomelike functions and, depending on the strain background, they can be larger than acidocalcisomes described e.g. in protists. We have elaborated on this point in the introduction of the revised version.

**Reviewer #3 (Public review):**
Bru et al. investigated how inorganic phosphate (Pi) is buffered in cells using *S. cerevisiae* as a model. Pi is stored in cells in the form of polyphosphates in acidocalcisomes. In *S. cerevisiae*, the vacuole, which is the yeast lysosome, also fulfills the function of Pi storage organelle. Therefore, yeast is an ideal system to study Pi storage and mobilization.They can recapitulate in their previously established system, using isolated yeast vacuoles, findings from their own and other groups. They integrate the available data and propose a working model of feedback loops to control the level of Pi on the cellular level.This is a solid study, in which the biological significance of their findings is not entirely clear. The data analysis and statistical significance need to be improved and included, respectively. The manuscript would have benefited from rigorously testing the model, which would also have increased the impact of the study.

It is not clear to us what the reviewer would see as a more rigorous test of the model.

**Reviewer #1 (Recommendations for the authors):**
(1) Figure 2: Why do the authors label the blue curve in A and B as BY and in C and D as WT? Is this a different genetic background they used here? This should be specified in the legend.

No, it is the same background. The figures had been reshuffled before submission and we overlooked to replace "BY" by "WT". This has been corrected. Now we consistently use WT in all figures

(2) Figure 4 has different scaling for the two panels, which should be labeled as A and B. I am aware that the authors do this for comparison, but it is rather confusing at first glance. I recommend having them at the same scale.

We chose this representation on two separate scales because this figure shall primarily illustrate that the shift between pho91Δ and WT curves vanishes in the presence of IP7. We now highlight in the figure legend that the scales are different to avoid confusion.

(3) Figure 8: I would appreciate a model with normal and low Pi concentrations in comparison, as this is what the authors worked out.

We have modified the figure. It now compares Pi-rich and Pi-limited scenarios.

(4) Minor issue: Wouldn't it make more sense to show the molar concentration in the Figures rather than the nmol of Pi/ug of protein? I am aware that this would require information on the vacuole volume rather than the reaction volume, and the authors do this calculation later on.

It depends. We often chose this representation because it illustrates the price to pay (metabolic input in terms of protein that must be dedicated to this task) to sequester a certain quantity of P^i^. But, as we provide the corresponding P^i^ concentration in the text, this information is accessible to the reader, too.

**Reviewer #2 (Recommendations for the authors):**
As stated above in the weaknesses section, while functional parallels exist, canonical acidocalcisomes are structurally and chemically distinct, typically smaller, electron-dense, and enriched with cations. Whereas yeast vacuoles are larger, multifunctional organelles with well-established roles beyond phosphate storage. Explicitly addressing these differences would strengthen the comparative framework and prevent potential confusion in interpreting the evolutionary relationships between these organelles.

We agree to some degree, which is the reason why we refer to vacuoles as acidocalcisome-like organelles. In fact, vacuoles share virtually all defining chemical traits of acidocalcisomes. They just have a second functional domain as hydrolytic, lysosome-like organelles. Given the plasticity of endo-lysosomal compartments, and acidocalcisomes belong to this group because of their biogenesis through the AP3 pathway, this is not shocking to us. But the reviewer's comment made us realize that it is better to explicitly address this point. We have added a section to the introduction to do this.

**Reviewer #3 (Recommendations for the authors):**
(1) Page 8: It is unclear why the authors only estimated the Pi concentration in wild-type vacuoles. This should also be done for vacuoles from other strains.

This information is inherent in Figure 2. PolyP hyperaccumulating strains show the same plateau as the wildtype, meaning that they also reach around 30 mM luminal Pi concentration, whereas vtc4Δ vacuoles reach only around 1/10th of that increase, indicating that they remain at 3 mM. We mention this now in the text.

(2) The attempts of the localization of Pho91 through tagging are not satisfactory. The author described different localizations for Pho91 depending on whether it was tagged on the N- or C-terminus or when Nterminally tagged and overexpressed using two strong promoters. While it is not uncommon that proteins show different localization patterns, depending on where the tag is inserted, it is possible that one of the tags would reflect the localization of the endogenous protein. There is an easy way to test this, in particular when Pho91 is endogenously tagged. pho91∆ has reported phenotypes such as abnormal vacuolar morphology or increased autophagy. They could also measure PI content in vacuoles. The authors could compare the phenotypes of the endogenously tagged strains with WT and a pho91∆ strain.

Indeed, the attempts to localise the protein through fluorescent tags are unsatisfactory, in our hands as in the hands of others. We would not have created a series of many different tagged versions (we present only a selection of these in the manuscript) if the creation of a faithful reporter for Pho91 localisation were so straightforward. Expression from the endogenous promoter yields quite low signals (which is why others have overexpressed their GFP fusion from strong promotors). But overexpression brings at least a significant part of the protein to the cell surface, where it can then function as Pi importer and suffice to restore much of the maximal Pi uptake capacity that genuine plasma membrane transporters provide and support normal growth of the cells (Wykoff & O’Shea, 2001). But the localisation pattern of Pho91-GFP, likewise overexpressed from a strong promotor, does not reflect this plasma membrane localisation (see the references that the reviewer mentioned under (3)). The published overexpressed GFP-fusions localise only to the vacuole, suggesting that even in this case the GFP tag may create an artefact. Therefore, we went through a large variety of Pho91 gene fusions, which led us to the conclusion that the protein is very sensitive to tags at both ends and that fusion proteins hence are unlikely to reliably report the correct location of the protein. Given this, we resorted to quantitative proteomics to clarify the issue. This quantitative experiment goes beyond previously published proteomics analyses that the reviewer mentions under (3), which found the protein in the vacuolar fraction but did not calculate the enrichment factors, which is crucial.

A strong phenotype of abnormal vacuolar morphology is not apparent in our cultures.

(3) Moreover, Pho91 has been identified as a component enriched in vacuolar-mitochondria contact sites (vCLAMP), and this localization was confirmed with GFP-Pho91 (PMID: 25026036). Likewise, PMID: 35175277 also detected Pho91 by mass spectrometry as a vacuolar protein and showed endogenously tagged GFP-Pho91 on the vacuole (co-staining with Vph1). The authors may request the strains from the authors of these papers and use them for their experiments. PMID: 17804816, the oldest of the three reports (from 2007) reports a GFP-Pho91 under either TEF or ADH promoter that localizes to the vacuole. They also showed that the fusion protein is functional. These and other experiments led them to conclude that Pho91 exports phosphate from the vacuolar lumen to the cytoplasma.

We have now included these references. As argued above, we have analysed also the strains from PMID17804816. The observed clear localisation of the fusion protein to vacuoles is only visible upon overexpression, not upon expression from the endogenous locus. Apparently also this construct is unlikely to report Pho91 localisation reliably (though, by chance, overexpression leads it to the correct location). Thus, we maintain our conclusion that C- or N-terminally GFP-tagged versions of Pho91 are unreliable tools for localising the protein.

(4) The impact of pho91∆ on Pho4-GFP nuclear localization is modest at best (increase from 5% of cells showing Pho4-GFP in the nucleus in WT vs 10% in pho91∆), and only somewhat stronger in ppn1∆/ppn2∆. This means 90% of pho91∆ cells do not respond, and Pho4-GFP stays cytoplasmic. It is unclear how the author can derive a meaningful conclusion from these data. Moreover, are these data really supporting the model, or do these data rather indicate that there are additional factors/pathways needed? What is the biological significance of the marginal increase from 5% to 10% of cells that would respond? What happens to the cells that cannot respond? Will they die or at least have a growth disadvantage? It would be useful to provide some functional studies.

We should have explained the nature of the assay better. The experiment exploits the fact that dividing yeast cells transiently fall into a state of Pi scarcity during S-phase. Since S-phase is less than a quarter of the cell cycle, only a small fraction of the cells transiently activates the PHO pathway. These cannot be well characterised by ensemble assays, but microscopy circumvents this background of the whole population and picks them up very clearly, allowing to quantify them. We have adapted the respective chapter in the results section to improve the description of this experiment.

(5) The quantification of the data is suboptimal, as in most assays the mean and standard error of the mean (SEM) are given. SEM is not really appropriate in these cases because it gives only the error of the mean and not of the entire data. Therefore, the standard deviation (SD) is needed, which reports on the variability of the data, and which is usually much larger than the SEM. Using the SD, would also allow the authors to do proper statistical analysis, which is missing entirely in this manuscript.

SEM also comprises the variability of the data. It is linked with the SD (SEM=SD/SQRT(n)), but SEM also considers the number of the experiments n. The main goal is to compare the means, and SEM is an appropriate and frequently used tool for this because it illustrates how well the arithmetic mean may estimate the true mean of the population. Therefore, we kept the SEM but have added tests of significance for the differences shown.

(6) Statistical testing in Figure 7 is essential as the effects are very small. Again, are these changes big enough for a biologically meaningful response? The authors should at least discuss this.

Our previous time course analyses of InsPP dynamics, performed under comparable conditions as in this study, showed that InsP8 decreases by around 50% in the first 30 min after transfer to Pi starvation (DOI: https://doi.org/10.7554/eLife.87956) and that this decline is already sufficient to trigger the PHO starvation program, as assessed by Pho4-GFP translocation into the nucleus. Thus, a 50% decrease, which is observed in ppn1 ppn2 mutants, is functionally significant. We have now also evaluated statistical significance in Fig. 7, which is given for the 50% reduction of InsP8 and 1-InsP7 in ppn1 ppn2.

Minor points:(1) There are a number of smaller edits (use of italic or better the absence thereof, lacking information in the reference list, and some typos).

Thank you. We have corrected those.

(2) The exact n should be given in the Figure legend.

Corrected.

(3) Page 8, line 8: it would be nice to have a picture of the wild-type vacuoles and what you measured.

We now present a sample image in the new Suppl. Fig. 1.

(4) PMID: 11779791 showed already that Pho91 cannot rescue the absence of the plasma membrane Pi transporters. This study should be at least cited.

This is not quite correct. The study that the reviewer mentions showed that Pho91 supports slower growth and the authors concluded that "A synthetic lethal phenotype was observed when (all) five phosphate transporters were inactivated...". We had cited the same group and the same first author, just using their later study (Wykoff et al., 2007) that had recapitulated the results from PMID11779791 and showed in addition quite good growth of the PHO91 expressing strain on YPD (Suppl. Fig. 2). We had obtained the strains from this group. In reproducing their experiments, we noticed that the growth of Pho91 that these authors had observed is due to incomplete repression of Pho84. They had overexpressed Pho84 from a galactose inducible promotor to generate a background with a regulatable Pi transporter. This trick allowed them to conveniently manipulate the strain and reduce (but not abolish) Pho84 expression by transferring the cells from galactose to glucose for their experiments. Therefore, we chose a more rigorous plasmid shuffling strategy to test the individual P_i_ transporter, which allows an assessment without the leaky background expression of Pho84 on glucose. In contrast to O'Shea and colleagues, we observed zero growth of a strain expressing only PHO91. We have revised the results section to make this discrepancy more evident and provide a better motivation for our experiment.

(5) It would be nice to see the actual data in Figure 6; not only a quantification.

We illustrate the phenotype of nuclear Pho4-GFP in panel A. Showing all the images necessary to appreciate the differences between the strains would require including many dozens of images into the figure, which would not be useful.